# Fleet-DAgger: Interactive Robot Fleet Learning with Scalable Human Supervision

**Ryan Hoque, Lawrence Yunliang Chen, Satvik Sharma,**
**Karthik Dharmarajan, Brijen Thananjeyan, Pieter Abbeel, Ken Goldberg**

University of California, Berkeley

**Abstract:**
Commercial and industrial deployments of robot fleets at Amazon, Nimble, Plus One, Waymo, and Zoox query remote human teleoperators when robots are at risk or unable to make task progress. With continual learning, interventions from the remote pool of humans can also be used to improve the robot fleet control policy over time. A central question is how to effectively allocate limited human attention. Prior work addresses this in the single-robot, single-human setting; we formalize the *Interactive Fleet Learning (IFL)* setting, in which multiple robots interactively query and learn from multiple human supervisors. We propose Return on Human Effort (ROHE) as a new metric and Fleet-DAgger, a family of IFL algorithms. We present an open-source IFL benchmark suite of GPU-accelerated Isaac Gym environments for standardized evaluation and development of IFL algorithms. We compare a novel Fleet-DAgger algorithm to 4 baselines with 100 robots in simulation. We also perform a physical block-pushing experiment with 4 ABB YuMi robot arms and 2 remote humans. Experiments suggest that the allocation of humans to robots significantly affects the performance of the fleet, and that the novel Fleet-DAgger algorithm can achieve up to $8.8\times$ higher ROHE than baselines. See https://tinyurl.com/fleet-dagger for supplemental material.

**Keywords:** Fleet Learning, Interactive Learning, Human-Robot Interaction

## 1 Introduction

Amazon, Nimble, Plus One, Waymo, and Zoox use remote human supervision of robot fleets in applications ranging from self-driving taxis to automated warehouse fulfillment [1, 2, 3, 4, 5]. These robots intermittently cede control during task execution to remote human supervisors for corrective interventions. The interventions take place either during learning, when they are used to improve the robot policy, or during execution, when the policy is no longer updated but robots can still request human assistance when needed to improve reliability. In the *continual learning* setting, these occur simultaneously: the robot policy has been deployed but continues to be updated indefinitely with additional intervention data. Furthermore, any individual robot can share its intervention data with the rest of the fleet. As opposed to robot swarms that must coordinate with each other to achieve a common objective, a robot *fleet* is a set of independent robots simultaneously executing the same control policy in parallel environments. We refer to the setting of a robot fleet learning via interactive requests for human supervision (see Figure 1) as *Interactive Fleet Learning (IFL)*.

Of central importance in IFL is the supervisor allocation problem: how should limited human supervision be allocated to robots in a manner that maximizes the throughput of the fleet? Prior work studies this in the single-robot, single-human case. A variety of interactive learning algorithms have been proposed that estimate quantities such as uncertainty [6], novelty [7, 8, 9], risk [8, 9], and predicted action discrepancy [10, 11]. However, it remains unclear which algorithms are the most effective when generalized to the multi-robot, multi-human case.

To this end, we formalize the IFL problem and present the IFL Benchmark (IFLB), a new open-source Python toolkit and benchmark for developing and evaluating human-to-robot allocation algorithms

---

AUTOLab at UC Berkeley. Correspondence to ryanhoque@berkeley.edu, goldberg@berkeley.edu.

6th Conference on Robot Learning (CoRL 2022), Auckland, New Zealand.

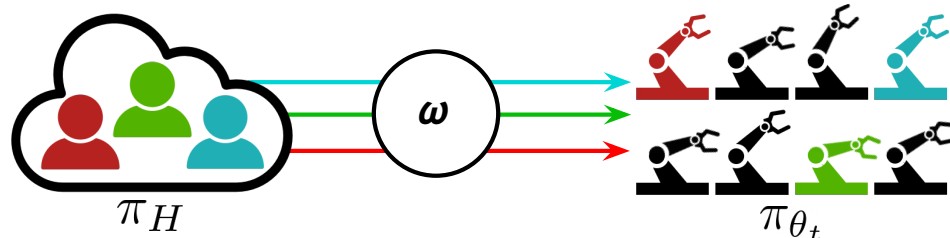

Figure 1: In the *Interactive Fleet Learning (IFL)* setting, a set of $M$ remote human supervisors are allocated to a fleet of $N$ robots ($N \gg M$) with a robot-gated allocation policy $\omega$. The humans share control policy $\pi_H$ and the robot fleet shares control policy $\pi_{\theta_t}$, which learns from new human intervention data over time (Section 3).

for fleet learning. The IFLB includes environments from Isaac Gym [12], which enabled efficient simulation of thousands of learning robots for the first time in 2021. This paper makes the following contributions: (1) the first formalism for multi-robot, multi-human interactive learning, (2) the Return on Human Effort (ROHE) metric for evaluating IFL algorithms, (3) the IFLB, an open-source software benchmark and toolkit for IFL algorithms with 3 Isaac Gym environments for complex robotic tasks, (4) Fleet-DAgger, a novel family of IFL algorithms for supervisor allocation, (5) results from large-scale simulation experiments with a fleet of 100 robots, and (6) real robot results with 4 physical robot arms and 2 human supervisors providing teleoperation remotely over the Internet.

## 2    Related Work

### 2.1    Allocating Human Supervisors to Robots at Execution Time

For human-robot teams, deciding when to transfer control between robots and humans during execution is a widely studied topic in the literature of both sliding autonomy [13, 14, 15] and Human-Robot Interaction (HRI). In sliding autonomy, also known as adjustable autonomy [16, 17] or adaptive automation [18], humans and robots dynamically adjust their level of autonomy and transfer control to each other during execution [15, 18]. Since identifying which robot to assist in a large robot fleet can be overwhelming for a human operator [19, 20, 21, 22, 23], several strategies, such as using a cost-benefit analysis to decide whether to request operator assistance [13] and using an advising agent to filter robot requests [24], have been proposed to improve the performance of human-robot teams [22, 24, 25] and increase the number of robots that can be controlled [26], a quantity known as "fan-out" [27]. Other examples include user modeling [13, 14, 22, 25] and studying interaction modes [28] for better system and interface design [29, 30]. Zheng et al. [31] propose computing the estimated time until stopping for mobile robots and prioritizing robots accordingly.  Ji et al. [32] consider the setting where physical assistance is required to resume tasks for navigation robots and formalize single-human, multi-robot allocation as graph traversal. Dahiya et al. [33] formulate the problem of multi-human, multi-robot allocation during execution as a Restless Multi-Armed Bandit problem. Allocation of humans to robots has also been studied from the perspectives of queueing theory and scheduling theory [23, 34, 35, 36, 37, 38]. The vast majority of the human-robot teaming and queueing theory work, however, does not involve learning; the robot control policies are assumed to be fixed. In contrast, we study supervisor allocation during robot learning, where allocation affects not only human burden and task performance but also the efficiency of policy learning.

### 2.2    Single-Robot, Single-Human Interactive Learning

Imitation learning (IL) is a paradigm of robot learning in which a robot uses demonstrations from a human to initialize and/or improve its policy [39, 40, 41, 42, 43, 44]. However, learning from purely offline data often suffers from distribution shift [45, 46], as compounding approximation error leads to states that were not visited by the human. This can be mitigated with online data collection with algorithms such as Dataset Aggregation (DAgger) [45] and interactive imitation learning [47, 48, 49]. Human-gated interactive IL algorithms [50, 51, 52] require the human to monitor the robot learning process and decide when to take and cede control of the system. While intuitive, these approaches are not scalable to large fleets of robots or the long periods of time involved in continual learning, as humans cannot effectively focus on many robots simultaneously [21, 22, 23] and are prone to fatigue [53]. To reduce the burden on the supervisor, several robot-gated interactive IL algorithms such as SafeDAgger [10], EnsembleDAgger [6], LazyDAgger [11], and ThriftyDAgger [9] have been proposed, in which the robot actively solicits human interventions when certain criteria are met.

Interactive reinforcement learning (RL) [54, 55, 56, 57, 58] is another active area of research in which robots learn from both online human feedback and their own experience. However, these interactive learning algorithms are designed for and primarily studied in the single-robot, single-human setting. Other works related to single-robot interactive learning include task allocation [59]; in contrast, we focus on efficient robot learning of a single control policy.

## 2.3 Multi-Robot Interactive Learning

In this paper, we study allocation policies for multiple humans and multiple robots. While many existing works [60, 61, 62, 63, 64] have leveraged NVIDIA's Isaac Gym's [12] capability of parallel simulation to accelerate reinforcement learning with multiple robots, these approaches do not involve human supervision. The work that is closest to ours is by Swamy et al. [65], who study the multi-robot, single-human problem of allocating the attention of one human operator during robot fleet learning. They propose to learn an internal model of human preferences as a human supervises a small fleet of 4 robots and use this model to assist the human in supervising a larger fleet of 12 robots. While this approach mitigates the scaling issue in human-gated interactive IL, even a small fleet of robots can be difficult for a single human supervisor to simultaneously monitor and control.

To the best of our knowledge, this work is the first to formalize and study multi-robot, multi-human interactive learning. This problem setting poses unique challenges, especially as the size of the fleet grows large relative to the number of humans, as each human allocation affects both the robot that receives supervision and the robots that do not receive human attention.

## 3 Interactive Fleet Learning Problem Formulation

We consider a fleet of $N$ robots operating in parallel as a set of $N$ independent Markov decision processes (MDPs) $\{\mathcal{M}_i\}_{i=1}^N$ specified by the tuple $(\mathcal{S}, \mathcal{A}, p, r, \gamma, p_i^0)$ with the same state space $\mathcal{S}$, action space $\mathcal{A}$, unknown transition dynamics $p : \mathcal{S} \times \mathcal{A} \times \mathcal{S} \to [0, 1]$, reward function $r : \mathcal{S} \times \mathcal{A} \to \mathbb{R}$, and discount factor $\gamma \in [0, 1)$, but potentially different initial state distributions $p_i^0$. We assume the MDPs have an identical indicator function $c(s) : \mathcal{S} \to \{0, 1\}$ that identifies which states $s \in \mathcal{S}$ violate a *constraint* in the MDP. States that violate MDP constraints are fault states from which the robot cannot make further progress. For instance, the robot may be stuck on the side of the road or have incurred hardware damage. We assume that the timesteps are synchronized across all robots and that they share the same non-stationary policy $\pi_{\theta_t} : \mathcal{S} \to \mathcal{A}$, parameterized by $\theta_t$ at each timestep $t$.

The collection of $\{\mathcal{M}_i\}_{i=1}^N$ can be reformulated as a single MDP $\mathcal{M} = (\mathcal{S}^N, \mathcal{A}^N, \bar{p}, \bar{r}, \gamma, \bar{p}^0)$, composed of vectorized states and actions of all robots in the fleet (denoted by bold font) and joint transition dynamics. In particular, $\mathbf{s} = (s_1, ..., s_N) \in \mathcal{S}^N$, $\mathbf{a} = (a_1, ..., a_N) \in \mathcal{A}^N$, $\bar{p}(\mathbf{s}^{t+1}|\mathbf{s}^t, \mathbf{a}^t) = \Pi_{i=1}^N p(s_i^{t+1}|s_i^t, a_i^t)$, $\bar{r}(\mathbf{s}, \mathbf{a}) = \Sigma_{i=1}^N r(s_i, a_i)$, and $\bar{p}^0 = \Pi_{i=1}^N p_i^0(s_i^0)$.

We assume that robots can query a set of $M \ll N$ human supervisors for assistance interactively (i.e., during execution of $\pi_{\theta_t}$). We assume that each human can help only one robot at a time and that all humans have the same policy $\pi_H : \mathcal{S} \to \mathcal{A}_H$, where $\mathcal{A}_H = \mathcal{A} \cup \{R\}$ and $R$ is a *hard reset*, an action that resets the MDP to the initial state distribution $s^0 \sim p_i^0$. As opposed to a *soft reset* that can be performed autonomously by the robot via a reset action $r \in \mathcal{A}$ (e.g., a new bin arrives in an assembly line), a *hard reset* requires human intervention due to constraint violation (i.e., entering some $s$ where $c(s) = 1$). A human assigned to a robot either performs hard reset $R$ (if $c(s) = 1$) or teleoperates the robot system with policy $\pi_H$ (if $c(s) = 0$). A hard reset $R$ takes $t_R$ timesteps to perform, and all other actions take 1 timestep.

Supervisor allocation (i.e., the assignment of humans to robots) is determined by an allocation policy

$$\omega : (\mathbf{s}^t, \pi_{\theta_t}, \boldsymbol{\alpha}^{t-1}, \mathbf{x}^t) \mapsto \boldsymbol{\alpha}^t \in \{0, 1\}^{N \times M} \quad \text{s.t.} \quad \sum_{j=1}^M \boldsymbol{\alpha}_{ij}^t \leq 1 \text{ and } \sum_{i=1}^N \boldsymbol{\alpha}_{ij}^t \leq 1 \quad \forall i, j, \quad (1)$$

where $\mathbf{s}^t$ are the current states for each of the robots, $\boldsymbol{\alpha}^t$ is an $N \times M$ binary matrix that indicates which robots will receive assistance from which human at the current timestep $t$, and $\mathbf{x}^t$ is an augmented state containing any auxiliary information for each robot, such as the type and duration of an ongoing intervention. Unlike Dahiya et al. [33] which studies execution-time allocation, the allocation policy $\omega$ here depends on the current robot policy $\pi_{\theta_t}$, which in turn affects the speed of the policy learning. While there are a variety of potential objectives to consider, e.g., minimizing constraint violations in a safety-critical environment, we define the IFL objective as *return on human*

*effort (ROHE)*:

$$\max_{\omega \in \Omega} \mathbb{E}_{\tau \sim p_{\omega, \theta_0}(\tau)} \left[ \frac{M}{N} \cdot \frac{\sum_{t=0}^{T} \bar{r}(\mathbf{s}^t, \mathbf{a}^t)}{1 + \sum_{t=0}^{T} \|\omega(\mathbf{s}^t, \pi_{\theta_t}, \boldsymbol{\alpha}^{t-1}, \mathbf{x}^t)\|_F^2} \right], \quad (2)$$

where $\Omega$ is the set of allocation policies, $T$ is the total amount of time the fleet operates (rather than the time horizon of an individual task execution), $\theta_0$ are the initial parameters of the robot policy, and $\|\cdot\|_F$ is the Frobenius norm. The objective is the expected ratio of the cumulative reward across all timesteps and all robots to the total amount of human time spent helping robots with allocation policy $\omega$, with a scaling factor to normalize for the number of robots and humans and an addition of 1 in the denominator for the degenerate case of zero human time. Intuitively, the ROHE measures the performance of the robot fleet normalized by the total human effort required to achieve this performance. We provide a more thorough derivation of the ROHE objective in Appendix 8.1.

Since human teleoperation with $\pi_H$ provides additional online data, this data can be used to update the robot policy $\pi_{\theta_t}$ with some policy update function $f$ (e.g., gradient descent):

$$\begin{cases} D^{t+1} \leftarrow D^t \cup D_H^t \text{ where } D_H^t := \{(s_i^t, \pi_H(s_i^t)) : \pi_H(s_i^t) \neq R \text{ and } \sum_{j=1}^{M} \boldsymbol{\alpha}_{ij}^t = 1\} \\ \pi_{\theta_{t+1}} \leftarrow f(\pi_{\theta_t}, D^{t+1}) \end{cases} \quad (3)$$

## 4 Interactive Fleet Learning Algorithms

### 4.1 Fleet-DAgger

Given the problem formulation above, we propose Fleet-DAgger, a family of IFL algorithms, where an *IFL algorithm* is a supervisor allocation strategy (i.e., it specifies an $\omega \in \Omega$ as defined in Section 3). The learning algorithm in Fleet-DAgger is interactive imitation learning with dataset aggregation from prior work [45, 50, 9]: its policy update function $f$ is supervised learning on $D^t$, which consists of all human data collected so far (Section 3). The novel component of each Fleet-DAgger algorithm is its supervisor allocation scheme based on unique *priority function* $\hat{p} : (s, \pi_{\theta_t}) \rightarrow [0, \infty)$ that indicates a priority score to assign to each robot based on its state $s$ and the current policy $\pi_{\theta_t}$, where, similar to scheduling theory, a higher value indicates a higher priority robot. To reduce thrashing [11, 9], Fleet-DAgger algorithms also specify $t_T$, the minimum time a human supervisor must spend teleoperating a robot.

Fleet-DAgger uses priority function $\hat{p}$ and $t_T$ to define an allocation $\omega$ as follows (see Appendix 8.2 for the full pseudocode). At each timestep $t$, Fleet-DAgger first scores all robots with $\hat{p}$ and sorts the robots by their priority values. If a human supervisor is currently performing hard reset action $R$ and $t_R$ timesteps have not elapsed, that human continues to help that robot. If a human is currently teleoperating a robot and the minimum $t_T$ timesteps have not elapsed, that human continues to teleoperate the robot. If a robot with a human supervisor continues to be high priority after the minimum intervention time ($t_R$ for a hard reset or $t_T$ for teleoperation) has elapsed, that human remains assigned to the robot. If a human is available to help a robot, the human is reassigned to the robot with the highest priority value that is currently unassisted. Finally, if a robot has priority $\hat{p}(\cdot) = 0$, it does not receive assistance even if a human is available.

### 4.2 Fleet-DAgger Algorithms

All algorithms below specify a unique priority function $\hat{p}$, which is synthesized with Fleet-DAgger as described in Section 4.1 to specify an allocation $\omega$. More details are available in the appendix.

**Constraint (C):** The Constraint baseline measures the performance of the robot fleet when only trained on offline human demonstrations (i.e., $\forall t, \pi_{\theta_t} = \pi_{\theta_0}$). At all timesteps $t$, this baseline gives priority $\hat{p}(\cdot) = 1$ for robots that have violated a constraint ($c(s_i^t) = 1$) and require a hard reset, and $\hat{p}(\cdot) = 0$ for all other robots. We refer to this as $C$-prioritization for Constraint. Thus, the robot fleet can only receive hard resets from human supervisors (no human teleoperation). Without $C$-prioritization, robots that require hard resets would remain indefinitely idle.

**Random:** This baseline simply assigns a random priority for each robot at each timestep. To control the total amount of human supervision, we introduce a threshold hyperparameter such that if a robot's priority value is below the threshold, its priority is set to zero and it will not request help.

**Fleet-EnsembleDAgger (U.C.):** This baseline adapts EnsembleDAgger [6] to the IFL setting. En-sembleDAgger uses the output variance among an ensemble of neural networks bootstrapped on

subsets of the training data as an estimate of epistemic uncertainty; accordingly, we define the robot priority for Fleet-EnsembleDAgger as ensemble variance. Since ensemble variance is designed for continuous action spaces, for environments with discrete action spaces we instead estimate the uncertainty with the Shannon entropy [66] among the outputs of a single classifier network. We refer to this priority function as $U$-prioritization for Uncertainty. Finally, since EnsembleDAgger was not designed for environments with constraint violations and idle robots will negatively affect the ROHE, we add $C$-prioritization for a more fair comparison. Specifically, given an uncertainty threshold value, robots with uncertainty above threshold are prioritized first in order of their uncertainty (U), followed by constraint-violating robots (C).

**Fleet-ThriftyDAgger (U.G.C.):** This baseline adapts the ThriftyDAgger algorithm [9] to the IFL setting. ThriftyDAgger uses a synthesis of uncertainty (which we refer to as the $U$-prioritization value) and the probability of task failure ($G$-prioritization, estimated with a Goal critic Q-function) to query a human for supervision. Since Fleet-DAgger requires a single metric by which to compare different robots, we adapt ThriftyDAgger to the fleet setting by calculating a linear combination of the $U$ value and $G$ value after normalizing each value with running estimates of their means and standard deviations. As in [9], we pretrain the goal critic on an offline dataset of human and robot task execution. Similar to Fleet-EnsembleDAgger, we first prioritize by the combined uncertainty-goal value above a parameterized threshold, followed by $C$-prioritization.

**Constraint-Uncertainty-Risk (C.U.R.):** Here we propose a novel Fleet-DAgger algorithm. As the name suggests, C.U.R. does $C$-prioritization, followed by $U$-prioritization, followed by $R$-prioritization. $R$ stands for Risk, which we define as the probability of constraint violation. Intuitively, idle robots should be reset in order to continue making progress, uncertain robots should receive more human supervision in areas with little to no reference behavior to imitate, and robots at risk should request human teleoperation to safety before an expensive hard reset. As in [67], we estimate the probability of constraint violation with a safety critic Q-function and initialize the safety critic on an offline dataset of constraint violations. C.U.R. also prioritizes differently at the beginning of execution for a parameterized length of time, during which constraint violations are assigned *zero priority* rather than high priority. Here, the intuition is that rather than attending to hard resets for an initially low-performing policy, human intervention should instead be spent on valuable teleoperation data that can improve the robot policy. Hence, during the initial period, constraint-violating robots remain idle and human attention is allocated to the teleoperation of a smaller number of robots.

## 5 Interactive Fleet Learning Benchmark

While many algorithms have been proposed for interactive learning [11, 6, 9, 10], to our knowledge there exists no unified benchmark for evaluating them. To facilitate reproducibility and standardized evaluation for IFL algorithms, we introduce the Interactive Fleet Learning Benchmark (IFLB). The IFLB is an open-source Python implementation of IFL with a suite of simulation environments and a modular software architecture for rapid prototyping and evaluation of new IFL algorithms.

### 5.1 Environments

The IFLB is built on top of NVIDIA Isaac Gym [12], a highly optimized software platform for end-to-end GPU-accelerated robot learning released in 2021, without which the simulation of hundreds of learning robots would be computationally intractable. The IFLB can run efficiently on a single GPU and currently supports the following 3 Isaac Gym environments with high-dimensional continuous state and action spaces (see Figure 2): (1) **Humanoid**, a bipedal legged locomotion task from OpenAI Gym [68], (2) **Anymal**, a quadruped legged locomotion task with the ANYmal robot by ANYbotics, and (3) **AllegroHand**, a task involving dexterous manipulation of a cube with a 4-finger Allegro Hand by Wonik Robotics. Constraint violation is defined as (1) the humanoid falling down, (2) the ANYmal falling down on its torso or knees, and (3) dropping the cube from the hand, respectively. End users can also add their own custom Isaac Gym environments with ease.

### 5.2 Software Architecture

The IFLB defines 3 interfaces for the development of IFL algorithms: (1) agents, (2) supervisors, and (3) allocations. An *agent* is an implementation of the robot fleet policy $\pi_{\theta_t}$ (Section 3), such as an IL or RL agent. A *supervisor* is an implementation of the supervisor policy $\pi_H$ (Section 3), such as a fully trained RL agent, a model-based planner, or a teleoperation interface for remote human supervisors. Lastly, an *allocation* is an implementation of the priority function $\hat{p}$ (Section 4), such as

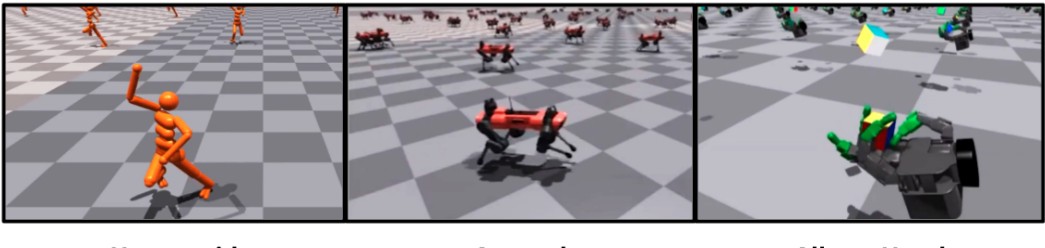

Figure 2: Isaac Gym benchmark environments in the IFLB.

C.U.R. priority or ThriftyDAgger priority. For reference, the IFLB includes an imitation learning agent, a fully trained RL supervisor using Isaac Gym's reference PPO [69] implementation, and all allocations from Section 4, which we use in our experiments. Users of the IFLB can flexibly implement their own IFL algorithms by defining new agents, supervisors, and allocations.

Given an agent, supervisor, allocation, and environment, the IFLB runs Fleet-DAgger as described in Section 4.1. IFLB allows flexible command line configuration of all parameters of the experiment (e.g., $t_T$, $t_R$, $N$, $M$) as well as the parameters of the agent, supervisor, and allocation. If desired, the code can also be modified to support families of IFL algorithms other than Fleet-DAgger. The benchmark is available open-source at https://github.com/BerkeleyAutomation/ifl_benchmark.

## 6 Experiments

### 6.1 Metrics

Throughout online training, we measure four metrics at each timestep $t$: (1) the cumulative number of successful task completions across the fleet and up to time $t$; (2) cumulative hard resets (i.e., constraint violations); (3) cumulative idle time, i.e., how long robots spend idle in constraint-violating states waiting for hard resets; and (4) the return on human effort (ROHE, Equation 2), where reward is a sparse $r \in \{0, 1\}$ for successful task completion and cumulative human time is measured in hundreds of timesteps. For the Humanoid and Anymal locomotion environments, success is defined as reaching the episode horizon without constraint violation and with reward of at least 95% of that of the supervisor policy. For the goal-conditioned tasks, i.e., AllegroHand and the physical block-pushing task, success is defined by reaching the goal state.

### 6.2 IFLB Simulation Experiments

**Experimental Setup:** We evaluate all Fleet-DAgger algorithms in the 3 benchmark simulation environments: Humanoid, Anymal, and AllegroHand. We use reinforcement learning agents fully trained with PPO [69] as the algorithmic supervisor $\pi_H$. We initialize the robot policy $\pi_{\theta_0}$ with behavior cloning on an offline dataset of 5000 state-action pairs. For a fair comparison, the Constraint baseline is given additional offline data equal to the average amount of human time solicited by C.U.R. by operation time boundary $T$. The Random baseline's priority threshold is set such that in expectation, it reaches the average amount of human time solicited by C.U.R. by time $T$. Since Fleet-ThriftyDAgger requires a goal-conditioned task, it is only evaluated on AllegroHand. All training runs are executed with $N = 100$ robots, $M = 10$ humans, $t_T = 5$, $t_R = 5$, and operation time $T = 10000$, and are averaged over 3 random seeds. In the appendix, we provide additional experiments including ablation studies on each component of the C.U.R. algorithm and an analysis of hyperparameter sensitivity to the number of humans $M$, minimum intervention time $t_T$, and hard reset time $t_R$. The IFLB code provides instructions for reproducing results.

**Results:** We plot results in Figure 3. First, we observe that the choice of IFL algorithm has a significant impact on all metrics in all environments, indicating that allocation matters in the IFL setting. We also observe that the robot fleet achieves a higher throughput (number of cumulative task successes) with C.U.R. allocation than baselines in all environments at all times. C.U.R. also attains a higher ROHE, indicating more efficient use of human supervision. An increase in ROHE over time signifies that the improvement in the robot policy $\pi_{\theta_t}$ outpaces cumulative human supervision, indicating that the IFL algorithms learn not only where to allocate humans but also when to *stop* requesting unnecessary supervision. C.U.R. also incurs fewer hard resets than baselines, especially

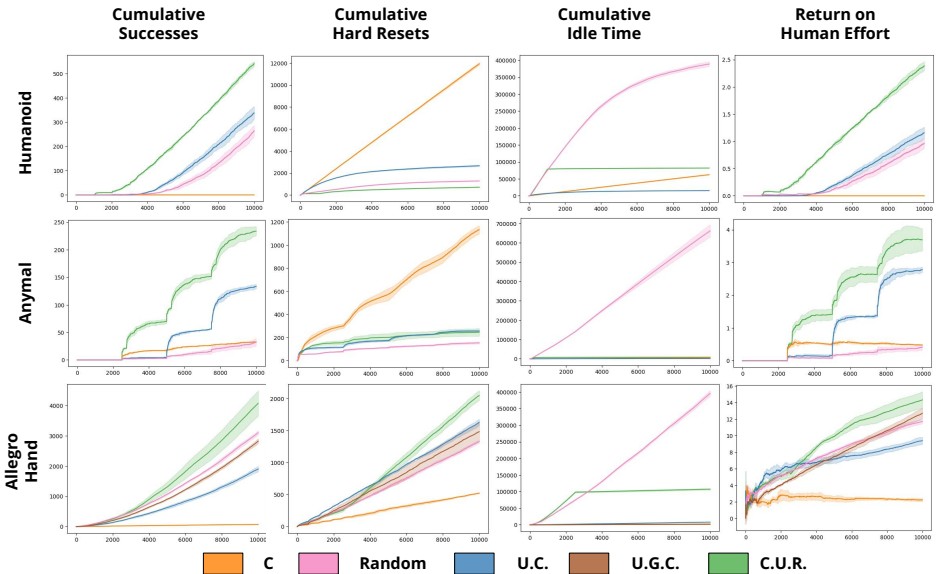

Figure 3: Simulation results in the IFLB with $N = 100$ robots and $M = 10$ human supervisors, where the $x$-axis is timesteps from 0 to $T = 10,000$. Shading indicates 1 standard deviation. The C.U.R. algorithm outperforms all baselines on all environments in terms of ROHE and cumulative successes. (Note that the shape of the Anymal curves is due to its success classification, episode horizon of 2500, and low hard resets.)

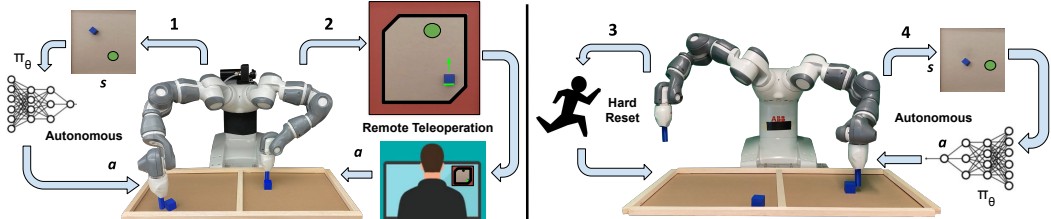

Figure 4: **Physical Task Setup:** an example timestep $t$ in the physical experiment with 2 humans and 4 independent identical robot arms each executing the block pushing task. **Robot 1** queries robot policy $\pi_{\theta_t}$ for an action given an overhead image of the workspace and executes it in the workspace. **Robot 2** is teleoperated by a remote Human 1, where the human views the overhead image and specifies a pushing action through a user interface. The red region at the edges of the workspace are constraint violation regions. Human 2 is performing a physical hard reset for **Robot 3**, which has violated a constraint in a previous timestep. **Robot 4** autonomously executes the same robot policy as that of Robot 1 on its own state.

Constraint, which must constantly hard reset robots with a low-performing offline policy. For AllegroHand, however, C.U.R. incurs higher hard resets and a smaller ROHE margin over baselines. We hypothesize that since the task is too challenging to execute without human supervision in the given fleet operation time, prioritizing hard resets ironically only gives the robots additional opportunities to violate constraints. We also see that $C$-prioritization effectively eliminates idle time; C.U.R. idle time flattens out after the initial period without $C$-prioritization.

## 6.3 Physical Block-Pushing Experiment

**Experimental Setup:** Finally, we evaluate Fleet-DAgger in a physical block-pushing experiment with $N = 4$ ABB YuMi robot arms and $M = 2$ human supervisors. Each robot arm has an identical setup for the block-pushing task that consists of a square wooden workspace, a small blue cube, and a cylindrical end-effector. See Figure 4 for the hardware setup. Constraint violation occurs when the cube hits the boundary or has moved into regions out of reach for the end-effector at two opposite corners of the workspace. The objective of each robot is to reach a goal position randomly sampled from the allowable region of the workspace. At each timestep, the robot chooses one of four discrete pushing actions corresponding to pushing each of the four vertical faces of the cube orthogonally by a fixed distance. The robot policy takes an overhead image observation of the cube in the workspace

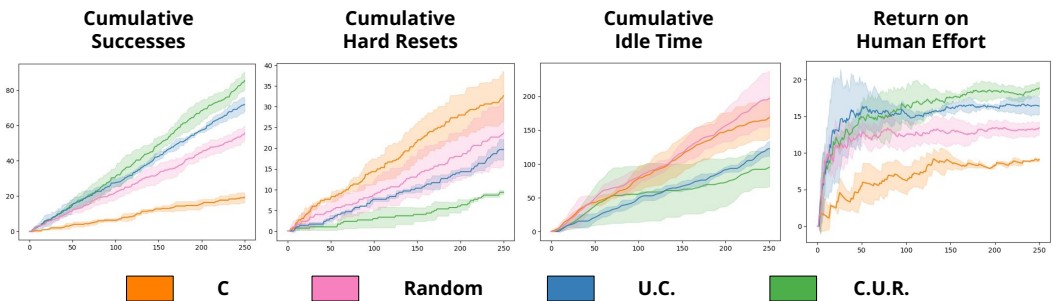

Figure 5: Physical results for the block-pushing task with 4 robots and 2 humans, where the $x$-axis is timesteps. C.U.R. achieves higher ROHE and cumulative successes as well as lower cumulative hard resets and idle time. Shading indicates 1 standard deviation.

with the goal programatically generated in the image. Hard resets are physical adjustments of the cube, while teleoperation is performed over the Internet by a remote human supervisor, who specifies one of the 4 pushing actions via a keyboard interface. We set $t_T = 3$, $t_R = 5$, and $T = 250$ for a total of $4 \times 250$ pushing actions per trial and run each algorithm with 3 random seeds. All algorithms are initialized with an offline dataset of 3750 image-action pairs (375 samples with $10\times$ data augmentation).

**Results:** We plot results in Figure 5. We observe that the C.U.R. algorithm achieves higher ROHE, higher cumulative successes, lower hard resets, and lower idle time than baselines, albeit by a small margin. Results suggest that (1) training an accurate safety critic is more difficult in high-dimensional image space, leading to a smaller gap between C.U.R. and U.C. (i.e., Fleet-EnsembleDAgger), and (2) $U$-prioritization in its current form is less suitable for real-world multimodal human supervisors than it is for deterministic algorithmic supervisors, resulting in a smaller increase in ROHE over time. Since a human may arbitrarily choose one of multiple equally suitable actions, high robot uncertainty over these actions does not necessarily translate to a need for human supervision.

# 7   Limitations and Future Work

The IFL formulation has a number of modeling assumptions that limit its generality. (1) The human supervisors are homogeneous, (2) all robots operate in the same state and action space, (3) all robots are independent and do not coordinate with each other, (4) humans have perfect situational awareness [23] and can move to different robots without any switching latency, (5) hard reset time is constant, and (6) timesteps are synchronous without network latency or other communication issues [70]. In terms of experiments, the simulations have algorithmic rather than human supervision, and the physical task is relatively straightforward with discrete planar actions.

In future work, we will work on removing the assumptions above: learning from nonhomogeneous supervisors [71] and collecting data from large fleets with limited network bandwidth [72], for instance. We will also study reinforcement learning algorithms for IFL and run more large-scale physical experiments. We hope that other robotics researchers will develop their own IFL algorithms and evaluate them using the benchmark toolkit to accelerate progress.

**Acknowledgments**

This research was performed at the AUTOLab at UC Berkeley in affiliation with the Berkeley AI Research (BAIR) Lab and the CITRIS "People and Robots" (CPAR) Initiative. The authors were supported in part by donations from Google, Siemens, Toyota Research Institute, and Autodesk and by equipment grants from PhotoNeo, NVidia, and Intuitive Surgical. Any opinions, findings, and conclusions or recommendations expressed in this material are those of the author(s) and do not necessarily reflect the views of the sponsors. We thank our colleagues who provided helpful feedback, code, and suggestions, especially Ashwin Balakrishna, Alejandro Escontrela, and Simeon Adebola.

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
