# OpenReview forum: "Fleet-DAgger: Interactive Robot Fleet Learning with Scalable Human Supervision"
_robot-learning.org/CoRL/2022/Conference — CoRL 2022 Oral_

### Official Review · Reviewer_3zng · 2022-07-16

**Originality:** Very Good
**Technical Quality:** Good
**Clarity Of Presentation:** Very Good
**Impact:** 4

**Recommendation:**

Weak Accept: I recommend accepting the paper, but will not argue for my recommendation if the majority of other reviewers have a different opinion.

**Summary:**

This paper presents interactive fleet learning, in which multiple robots interactively query and learn from multiple humans. The new setting applies DAGGER on a fleet of robots and is tested in simulators and the real world. The key contribution is a novel objective to measure return on human efforts during training and different ways of assigning priorities based on constraints and uncertainty.

**Issues:**

Not sure if I understand ROHE correctly. The definition seems to be dependent on M and N. If we increase N, then the numerator would get bigger since it is a sum instead of a mean, but the denominator might not change. If we increase M, the denominator would become large (if no idle humans), but the behavior on the numerator is harder to analyze because more humans might increase the overall training performance. Overall the ratio seems to complicate the definition.

More details on how to pretrain goal critics and safety critics are very helpful.

For the simulation experiment, why is CUR having a higher idle time (isn’t it a bad thing), and why in the real world it is not the case.
More results on test performance and visualization are helpful.

Suggest moving figure 3 to earlier part of the paper.

Variance can be useful to provide for multiple runs on the training process, especially with humans in the loop.


**Quality Of The Limitations Section:**

Limitations are addressed clearly

**Reviewer Expertise:**

4: The reviewer is confident but not absolutely certain that the evaluation is correct

**Robotics Focus:**

Sufficient demonstration on hardware

**Strengths And Weaknesses:**

Strength:

The paper considers a very practical problem for robot fleets in the industry where minimizing human efforts and maximizing throughput is of critical importance.

Compared to related works on execution time, fleet-dagger focuses on training time, and it is more scalable as a robot-gated algorithm.

The details on the formulation are novel and meaningful, such as hard reset time R takes more than 1 timestep, and T_t for minimum operation time. The consideration of constraints is also very practical (such as dropping the box in the allegro hand example).

The software and protocol are also contributions.

Weakness:

The multi-human and multi-robot setting is novel, but the humans are homogeneous and robots share a single policy for the same task, which is limited and not much different from single robot-human setting except the parallel implementation (parallel MDPs). Additionally, there might be a communication bottleneck to share all data among a large fleet.

The algorithm baselines are mostly from previous works and CUR seems to be a combination with additional constraint prediction. Since these functions can be trained, assigning in order based on this score seems not that difficult. It would be interesting to consider the execution-time human-robot allocation methods as well. Moreover, some methods require pretrain goal critic or safety critic. One question is how hard it is in practice to train such an additional safety function, compared to the policy learning.

Fleet-dagger only focuses on the interactive learning process. But it would be very important to evaluate the trained policy as well. Does the final trained policy differ among these different methods, or only the training process is sped up? Are there some policy visualizations or generalization analyses?

Since the title is fleet robotics, it would be great to have more robots and humans (beyond 4 arms and 2 robots) with more complex tasks (beyond pushing) on the real-world experiments.


**Summary Of Recommendation:**

This paper presents a well-thought and clearly written formulation for an important real-world robotic problem. The protocol and codebase are also very useful for the community in general. There are still several weaknesses regarding formulation and algorithms and would love to hear about the response from the authors. Overall I recommend acceptance of this paper.

---

> ### Author Response · Authors · 2022-08-20
> **Response to Reviewer 3zng**
>
> **We thank Reviewer 3zng for their feedback and positive comment that** “The paper considers a very practical problem for robot fleets in the industry where minimizing human efforts and maximizing throughput is of critical importance.” **We address each concern below. We have attached an updated draft of the paper + appendix to this comment reply with all the changes indicated in the responses below highlighted in blue.**
>
> There might be a communication bottleneck to share all data among a large fleet.
>
> **Good point, we have added this limitation to the limitations section.**
>
> The multi-human and multi-robot setting is novel, but the humans are homogeneous and robots share a single policy for the same task, which is limited and not much different from single robot-human setting except the parallel implementation (parallel MDPs).
>
> **The multi-human, multi-robot setting differs from the single-human, single-robot case because each assignment affects the learning and execution of the entire fleet, and because the ratio of humans to robots is not 1:1. Additionally, we have updated the formalism to allow different initial state distributions for each MDP. Note that although the fleet must share the same policy, the “task” can be defined quite broadly; for instance, a goal-conditioned policy with a language prompt can be leveraged for many different tasks.**
>
> Since these functions can be trained, assigning in order based on this score seems not that difficult. It would be interesting to consider the execution-time human-robot allocation methods as well.
>
> **We agree that the execution-time allocation methods would be interesting to study in this setting, and hope to extend the IFL benchmark to a library of allocation implementations in the future. Note, however, that some of the most relevant papers in the execution-time literature are restricted to mobile robot navigation (e.g., Zheng et al [30]) or modeling non-homogeneous humans (Sellner et al. [12]). The current set of IFL algorithms are more general than only navigation, and the IFL setting currently assumes homogeneous humans.**
>
> Moreover, some methods require pretrain goal critic or safety critic. One question is how hard it is in practice to train such an additional safety function, compared to the policy learning. … More details on how to pretrain goal critics and safety critics are very helpful.
>
> **We have added details on training safety and goal critics in practice to Appendix 8.3.2.**
>
> Does the final trained policy differ among these different methods, or only the training process is sped up? Are there some policy visualizations or generalization analyses? … More results on test performance and visualization are helpful.
>
> **We have added evaluations for the final trained policies in Table 2 of the Appendix. Results indicate that the trained policies do indeed vary in performance. We have also added policy visualizations as requested to the anonymous project website (https://sites.google.com/view/fleet-dagger), as the files are too large to attach to the portal.**
>
> Since the title is fleet robotics, it would be great to have more robots and humans (beyond 4 arms and 2 robots) with more complex tasks (beyond pushing) on the real-world experiments.
>
> **We agree that it will be important to evaluate with more physical robots and humans and more complex tasks. Unfortunately for now we are constrained by cost and other resources to 4 real robots, but are very interested in experimenting with larger physical fleets in the future.**
>
> Not sure if I understand ROHE correctly. The definition seems to be dependent on M and N. If we increase N, then the numerator would get bigger since it is a sum instead of a mean, but the denominator might not change. If we increase M, the denominator would become large (if no idle humans), but the behavior on the numerator is harder to analyze because more humans might increase the overall training performance. Overall the ratio seems to complicate the definition.
>
> **Thank you for asking for this clarification.  Indeed, increasing N increases the numerator and increasing M increases the denominator (and possibly the numerator). As such, we have added a M/N scaling factor to the ROHE objective for normalization. Regarding the ratio, we have also considered formulating the objective as an additive weighted penalty (numerator - lambda * denominator) in order to decompose over time, but this inherits other undesirable properties—namely, the value would compound over time.**

---

> ### Author Response · Authors · 2022-08-20
> **Response to Reviewer 3zng (Part 2)**
>
> **(continuing reply due to space constraints)**
>
> For the simulation experiment, why is CUR having a higher idle time (isn’t it a bad thing), and why in the real world it is not the case.
>
> **C.U.R. has a high idle time due to the initial period during which constraint-violating robots have *zero priority* instead of high priority (in order to prioritize teleoperation data). As a result, a “cliff” is visible in the idle time plots, before which idle time climbs for the idle robots and after which is effectively eliminated (once constraints are highly prioritized). In the real experiments, the fleet is much smaller (4 robots) resulting in lower cumulative idle time of the fleet and therefore a less visible trend.**
>
> Variance can be useful to provide for multiple runs on the training process, especially with humans in the loop.
>
> **Error bars are included in Fig 2 to indicate variance, and in the revision we have also added error bars to the physical experiments.**

---

### Official Review · Reviewer_45kd · 2022-07-28

**Originality:** Good
**Technical Quality:** Very Good
**Clarity Of Presentation:** Excellent
**Impact:** 2

**Recommendation:**

Strong Accept: I recommend accepting the paper and will argue for my recommendation even if other reviewers hold a different opinion.

**Summary:**

Review for Fleet-DAgger: Interactive Robot Fleet Learning with Scalable Human Supervision
The paper studies the problem of allocating multiple human teleoperators to a fleet of robots. In a formal framework called Interactive Fleet Learning (IFL), robots query the humans to improve their behaviour. Robots are not acting as swarm, but rather do the same tasks in parallel and can share their data obtained from interacting with humans among each other. Finally, the setting is continual learning, human help is not only available in training but also during execution to further improve the robot’s policies. The authors provide a benchmark toolkit for IFL, as well as a novel IFL algorithm adapting DAgger. The work is evaluated in simulation and in hardware experiments.


**Issues:**

All issues have been addressed during the rebuttal phase.

**Quality Of The Limitations Section:**

Limitations are addressed clearly

**Reviewer Expertise:**

3: The reviewer is fairly confident that the evaluation is correct

**Robotics Focus:**

Sufficient demonstration on hardware

**Strengths And Weaknesses:**

Overall, the paper is very well written, clearly structured and studies a relevant problem to the robot learning community. Related work is reviewed in great detail, and the problem statement is thorough. The paper makes two main contributions (IFL formalism and the C.U.R. algorithm). The simulation experiments show strong results. However, I have a technical question on the objective and some concerns about the hardware experiments.Please see detailed comments below:

Major comments:

•	The Fleet-Dagger family that unifies different algorithms is nice and provides a useful building block for designing or improving IFL algorithms

•	resolved -- About the objective in eq (2): Isn’t this objective trivially maximized by having the humans spent 0 time teleoperating? This should be clarified.

•	resolved -- Physical Experiment in Section 6.3: How many trials were carried out for this? How were human participants selected? Why are there no error bands in Figure 4? Especially with the smaller margin between algorithms, error bands / SD would be very helpful to evaluate if there is sufficient performance difference to draw any conclusions.

Minor comments:
•	resolved -- Fig 2: The step like behaviour for CUR and Fleet-Ensemble-Dagger in the Anymal experiment are peculiar. Is there any explanation for this?

•	resolved -- A few typos, e.g. line 207 misses a space before ‘In addition’

•	resolved -- Reference 2 is not well formatted

Limitation Section:

•	The limitation section is adequate.


**Summary Of Recommendation:**

The paper makes valuable contributions with a unified problem formulation, the family of Fleet-Dagger algorithms, the novel CUR algorithm and a strong simulation study in a new benchmark testbed. The work is evaluated in thorough simulations, and its practicality is demonstrated in a set of hardware experiments.

---

> ### Author Response · Authors · 2022-08-20
> **Response to Reviewer 45kd**
>
> **We thank Reviewer 45kd for their feedback and comment that** “the paper is very well written, clearly structured and studies a relevant problem.” **We address each concern below. We have attached an updated draft of the paper + appendix to this comment reply with all the changes indicated in the responses below highlighted in blue.**
>
> About the objective in eq (2): Isn’t this objective trivially maximized by having the humans spent 0 time teleoperating? This should be clarified.
>
> **Thanks for catching that! We’ve added a +1 to the denominator to remedy the problem you pointed out: human time = 0 would be a degenerate case that results in an infinite return.**
>
> Physical Experiment in Section 6.3: How many trials were carried out for this? How were human participants selected? Why are there no error bands in Figure 4? Especially with the smaller margin between algorithms, error bands / SD would be very helpful to evaluate if there is sufficient performance difference to draw any conclusions.
>
> **In our original submission we had results for only 1 trial per algorithm, as each full algorithm run takes a long time on a physical system. As a result we did not have error bands. Since then we have run new physical experiments with a square pusher (as opposed to a cylindrical one) to mitigate cube rotation and have run 3 trials for each algorithm (3 trials x 4 algorithms x 4 robots x 250 timesteps = 12000 actions). As suggested we have added the new results with error bars to Figure 4. In terms of human participants, since the physical experiments are very time-consuming, the teleoperation and hard resets were performed by the authors of the paper; we added this information to Appendix 8.3.3.**
>
> Fig 2: The step like behaviour for CUR and Fleet-Ensemble-Dagger in the Anymal experiment are peculiar. Is there any explanation for this?
>
> **We added this information to the figure caption. The reason for this is the way success is calculated in the Anymal environment: each robot is successful if it can reach 95% of expert reward. In the Anymal environment, to reach this the robot must locomote for nearly the full episode horizon of 2500. In addition, hard resets in Anymal are relatively rare (e.g., compared to Humanoid), so the whole fleet is more synchronized as to when they reach this reward threshold.**
>
> A few typos, e.g. line 207 misses a space before ‘In addition’. … Reference 2 is not well formatted.
>
> **Nice catch! Fixed.**

---

> > ### Comment · Reviewer_45kd · 2022-08-25
> > **Reply**
> >
> > I appreciate the effort the authors put into the revision of their paper, especially the new physical experiments.
> >
> > My concerns have been adequately addressed, thank you!

---

### Official Review · Reviewer_tye5 · 2022-07-30

**Originality:** Good
**Technical Quality:** Very Good
**Clarity Of Presentation:** Very Good
**Impact:** 4

**Recommendation:**

Strong Accept: I recommend accepting the paper and will argue for my recommendation even if other reviewers hold a different opinion.

**Summary:**

Fleet-DAgger addresses the following question. How can a fleet of independent robots, each operating the same policy, most efficiently learn a task, given the availability of a small number of humans who can temporarily take over control from the policy and perform interventions? The authors call this Interactive Fleet Learning (IFL).

The main contributions of the paper are as follows:
* A framework for doing IFL. The framework’s simulation code and results are open-sourced
* An allocation algorithm (CUR) for allocating supervision (human or otherwise) to robots
* A clear set of metrics by which to compare IFL algorithms, including return on human effort (ROHE)
* A detailed comparison of CUR with relevant baselines and an extensive set of ablations which shed light on the contributions of different components of the algorithm.
* An instantiation of IFL on a physical system consisting of 4 robots + 2 humans.


**Issues:**

* It would be helpful if the authors could unify the terminology between the baselines and the ablations. The C, U, R, UR, etc notation is really clear and it strikes me that all of the baselines, except for the random baseline, could be described in this way.
   * E.g. behavior cloning seems equivalent to C, Fleet-Ensemble DAgger seems equivalent to U.C, Fleet-Thrifty DAgger is a little different but could be described as UG.C
* Please could the authors calculate ROHE for BC
* Please could the authors clarify if other algorithms had a warm start
* Please could the authors clarify the cumulative success rate on the real robot
* I am curious what the data augmentations are (301). It would be helpful if this could be clarified in the appendix.
* Line 265 “we use reinforcement learning agents fully trained with PPO”. What does fully trained mean? It would be helpful if the scores per environment could be given for these trained agents. For example, how many cumulative successes would the supervisor PPO agent achieve in the same 10,000 timesteps.
* Line 163: What does “high priority” mean here. For example could it ever be that a human does not switch to assist an unassisted robot with a higher priority than the currently assisted robot?
* Line 97 “a small fleet of robots can be difficult for a single human supervisor to optimally control” It is not clear to me why scaling the number of supervisors would help here. More concretely it is not clear there is a difference between 4 robots and 1 human and 20 robots and 5 humans, since each has to operate 4 robots on average. Please could the authors clarify this.
* Appendix Algorithm 1: line 15, should this be N not M?


**Quality Of The Limitations Section:**

Limitations are addressed clearly

**Reviewer Expertise:**

3: The reviewer is fairly confident that the evaluation is correct

**Robotics Focus:**

Sufficient demonstration on hardware

**Strengths And Weaknesses:**

Strengths
* This work is both novel and relevant. Given that behavior cloning (BC), has been demonstrated to be an effective method for robot learning (e.g. What Matters in Learning from Offline Human Demonstrations for Robot Manipulation, Mandlekar et al, 2021) and the benefits of scaling up learning have been clearly demonstrated in other areas of ML, I can see the importance of research into effective and scalable fleet learning growing. Further I have not seen much work in fleet learning to date and so I think this work could be an important step in that direction. Finally, since human time is costly (e.g Back to Reality for Imitation Learning, Johns, 2021 (CoRL blue sky)), I think this work makes a valuable contribution by proposing a method which attempts to maximize return on human effort.
* The paper is well written throughout and the motivation is especially clear.
* The literature review appears thorough (although I am not an expert on fleet learning)
* The selection of baselines (4 different baselines) is thorough and relevant.
* I especially appreciate the detailed set of ablations in the supplementary materials which shed light on the different components of the CUR algorithm. The terminology (U, UR, CUR etc) is also really clear.
   * I appreciate that there are space constraints but it would be great to see a brief summary of the main conclusions from these ablations in the main paper.
* The experiment selection is thorough covering 4 tasks (3 in simulation and 1 on physical robots)
* The framework appears quite general and also reproducible. The authors open source both the code and results for simulation.
* Results are demonstrated on a fleet of physical robots (4 robots + 2 humans)
* The benefit of CUR over baselines in simulation appear to be substantial. However I have a few questions detailed below.

Weaknesses
* The warm start appears to have a significant impact on CUR performance (Figure 5) however it is not clear if the baselines were also allowed a warm start.
* ROHE is not calculated for BC. The authors say this is because it is “an offline algorithm” (line 256) however it is not clear to me why this justifies not calculating it. It appears that it could be calculated given the definition of ROHE and would be a valuable comparison.
* The benefit of CUR over baselines on the physical robot appears marginal
* The success rate on the physical robot seems very high for even the poorest performing algorithm. If I understood this correctly there are 250 timesteps and 4 robots making a maximum of 1000 cumulative successes (and also 1000 total actions across all robots and humans (line 300)) . The worst performing policy (Random) scored 500 cumulative successes implying that a robot succeeded at the task every 2 actions. This seems high.


**Summary Of Recommendation:**

In my opinion, this paper currently stands on the border of a weak and strong accept. The problem setting is both novel and relevant, the paper is well written, the experiments are well chosen and thorough. I think the work makes a valuable contribution to the community by establishing a framework for IFL, defining a useful set of metrics, establishing a set of baselines and shedding light on what factors affect learning effectiveness through a thorough set of experiments and ablations.

There are some issues with the paper which I highlight in the weaknesses and below. The main one being that I am currently unsure how much of an improvement CUR represents over existing algorithms because of (a) the benefit of the warm start for CUR and it is unclear if other algorithms use this and (b) not calculating ROHE for BC and c) the marginal benefit of CUR on the physical robot

However for me the strengths clearly outweigh the weaknesses and I don’t think the impact of this works rests on how well CUR performs.

---

> ### Author Response · Authors · 2022-08-20
> **Response to Reviewer tye5 (Part 1)**
>
> **We thank Reviewer tye5 for their feedback and note that** “This work is both novel and relevant” **and that the** “paper is well written throughout and the motivation is especially clear.”  **We address each concern below. We have attached an updated draft of the paper + appendix to this comment reply with all the changes indicated in the responses below highlighted in blue.**
>
> ROHE is not calculated for BC. The authors say this is because it is “an offline algorithm” (line 256) however it is not clear to me why this justifies not calculating it. It appears that it could be calculated given the definition of ROHE and would be a valuable comparison. … Please could the authors calculate ROHE for BC.
>
> **Thank you, that’s a great point, we have added ROHE for BC into the simulation and physical plots as requested.**
>
> The warm start appears to have a significant impact on CUR performance (Figure 5) however it is not clear if the baselines were also allowed a warm start. … Please could the authors clarify if other algorithms had a warm start.
>
> **Thanks for pointing this out. All algorithms actually do have a warm start in the sense that the robot policies are initialized via behavior cloning on an offline dataset before starting online training. What we called the “warmup period” for C.U.R. was an initial time period during which C.U.R. behaves differently from C.U.R. in a later time period. You can think of C.U.R. as 2 allocation strategies pasted together. The “warmup” term is misleading as there is no “warm start” occurring - it is part of the allocation algorithm defined by C.U.R. (that indeed may be playing a large role in performance, as you note), which doesn’t readily extend to the other baselines as they allocate differently. Accordingly, we have removed the term “warmup” from the text.**
>
> The benefit of CUR over baselines on the physical robot appears marginal.
>
> **We have run 3 new physical trials of each algorithm (3 trials x 4 algorithms x 4 robots x 250 steps = 12000 actions) with square pushers (as opposed to cylindrical ones from the original submission) to reduce cube rotation which allows us to add error bars to more clearly evaluate each algorithm. While U.C. and C.U.R. still perform similarly, the shaded regions for standard deviations do not overlap, suggesting that the benefits of C.U.R. over baselines are larger than previously reported.**
>
> The success rate on the physical robot seems very high for even the poorest performing algorithm. If I understood this correctly there are 250 timesteps and 4 robots making a maximum of 1000 cumulative successes (and also 1000 total actions across all robots and humans (line 300)) . The worst performing policy (Random) scored 500 cumulative successes implying that a robot succeeded at the task every 2 actions. This seems high. … Please could the authors clarify the cumulative success rate on the real robot.
>
> **Yes, thanks for pointing this out. The reason for this is the way success was calculated for the physical experiments in the original submission: any push in a direction that reduces distance to the goal was labeled 1 success, and successfully reaching the goal was labeled as 5 successes. This resulted in frequent success and high cumulative success numbers. To fix this, for the new experiments in the revision we instead calculate only goal-reaching as 1 success (and 0 successes for push direction), which makes cumulative success more sparse but more interpretable.**
>
> It would be helpful if the authors could unify the terminology between the baselines and the ablations. The C, U, R, UR, etc notation is really clear and it strikes me that all of the baselines, except for the random baseline, could be described in this way.
>
> **You are correct that Fleet-EnsembleDAgger can be labeled as UC and Behavior Cloning can be labeled as C. We’ve added the labels you suggest to the plots and Section 4.**
>
> I am curious what the data augmentations are (301). It would be helpful if this could be clarified in the appendix.
>
> **We have added these details in Appendix Section 8.3.3 as requested. They are primarily brightness augmentations as well as a small amount of noise.**
>
> Line 265 “we use reinforcement learning agents fully trained with PPO”. What does fully trained mean? It would be helpful if the scores per environment could be given for these trained agents. For example, how many cumulative successes would the supervisor PPO agent achieve in the same 10,000 timesteps.
>
> **Fully trained in this case means training for the full default length of training in NVIDIA’s IsaacGymEnvs reference implementation: https://github.com/NVIDIA-Omniverse/IsaacGymEnvs. The parameters can be found in the config files of the repository. For instance, Humanoid runs for 1000 epochs with 4096 parallel environments. As requested, we added the statistics for how the supervisors perform over 10000 steps in Table 2 of the appendix as “Expert.”**

---

> ### Author Response · Authors · 2022-08-20
> **Response to Reviewer tye5 (Part 2)**
>
> **(Continuing the previous reply due to space constraints)**
>
> Line 163: What does “high priority” mean here. For example could it ever be that a human does not switch to assist an unassisted robot with a higher priority than the currently assisted robot?
>
> **High priority here means that the robot will get some form of human assistance from the available pool of supervisors (where M is the number of humans). Here we indicate that if a robot that is currently receiving supervision will continue to receive it from some human in the next timestep, we might as well keep the human helping it in place. A human will always switch to a higher priority robot if the minimum intervention time for their current robot has elapsed and no one else is available to help the higher priority robot.**
>
> Line 97 “a small fleet of robots can be difficult for a single human supervisor to optimally control” It is not clear to me why scaling the number of supervisors would help here. More concretely it is not clear there is a difference between 4 robots and 1 human and 20 robots and 5 humans, since each has to operate 4 robots on average. Please could the authors clarify this.
>
> **You are correct that 4 robots: 1 human and 20 robots: 5 humans would be equally difficult. Here we are noting that in Swamy et al. [63], 1 human must monitor all 4 robots simultaneously in a “human-gated” manner where the human must determine which one needs help. In contrast, in our “robot-gated” setting the robots actively query for help, so the humans do not have to passively monitor robot execution and each human only helps one robot at a time. We have updated the text to better reflect this: “While this approach mitigates the scaling issue in human-gated interactive IL, even a small fleet of robots can be difficult for a single human supervisor to simultaneously monitor and control.”**

---

> > ### Comment · Reviewer_tye5 · 2022-08-27
> > **Thank you for your detailed response**
> >
> > Thank you for your detailed response and for your edits to the paper. The authors have addressed all my concerns and questions.
> >
> > Minor points
> > * It seems that the UGC results are missing from Figure 2 for the Anymal and Humanoid environments in the updated paper. Is this intended?
> > * I appreciate the updates to success labeling for real world experiments. It is much clearer now. It would be interesting to know what the maximum number of possible successes are given the number of robots and actions (4 x 250). This would give some indication of how close to optimal the supervisor allocation scheme is, although I note that the performance of the underlying PPO policy would be a confounding factor.

---

> > > ### Author Response · Authors · 2022-08-27
> > > **Response to Reviewer tye5's Response**
> > >
> > > Thank you Reviewer tye5, we are glad that you feel we have addressed your concerns!
> > >
> > > For the minor points:
> > >
> > > 1. Yes, UGC (Fleet-ThriftyDAgger) is missing from Anymal and Humanoid in the updated and original paper. This is intentional, as the goal critic from ThriftyDAgger that produces the G-value requires the task to be goal-conditioned (i.e., there is some set of goal states that define task success); see Line 268. Since AllegroHand is goal-conditioned (success = manipulating the cube into a desired state) and Anymal and Humanoid are not (success = locomotion without constraint violation and with sufficient reward accrued by the episode horizon), we can only evaluate UGC on the former.
> > >
> > > 2. From the demonstration data collected for the latest experiments, the expert achieves 67 successes in 500 timesteps for a single robot. Extrapolating to 1000 timesteps (4 x 250), we can estimate that the maximum number of possible successes is about 134. Factors contributing to the gap between the algorithms and this upper bound include suboptimality in supervisor allocation (as you say); the quality of the robot policy, especially earlier on in training (as you say); and only having 2 human supervisors for the 4 robots rather than 1 human per robot.

---

### Official Review · Reviewer_oMQS · 2022-07-31

**Originality:** Very Good
**Technical Quality:** Excellent
**Clarity Of Presentation:** Very Good
**Impact:** 4

**Recommendation:**

Strong Accept: I recommend accepting the paper and will argue for my recommendation even if other reviewers hold a different opinion.

**Summary:**

The authors formalize and provide research tools (ie, benchmark domains and metrics) for the important problem of "interactive fleet learning," in which in which a set of remote (human) operators interact with a fleet of deployed robots over time and the goal is for the fleet to use the interaction data to reduce its dependence upon the human operators. The authors additionally propose and provide experimental results for a set of algorithms that they refer to as Fleet-DAgger which serve to determine which humans should interact with which robots at what times. The authors find in experiments that the proposed "CUR" scheme leads to better fleet learning.

**Issues:**

(see weaknesses identified above)

**Quality Of The Limitations Section:**

Additional details required

**Reviewer Expertise:**

5: The reviewer is absolutely certain that the evaluation is correct and very familiar with the relevant literature

**Robotics Focus:**

Sufficient demonstration on hardware

**Strengths And Weaknesses:**

(+) The paper articulates well an important problem in the robot learning community, ie, that of interactive fleet learning (IFL). Moreover, the paper introduces a new—and seemingly valuable—set of benchmark tasks that instantiate this problem

(+) The paper proposed an intuitive and attractive "return on human effort" (RHOE) metric for IFL solutions.

(+) The proposed method for supervisor allocation, ie, CUR, performs well in a strong experimental evaluation.



(-) While I appreciate the overall definition of IFL in the introduction (ie, "a robot fleet learning via interactive requests for human supervision"), the Fleet-DAgger approach(es) presented in the paper focus on the particular sub-problem of "supervisor allocation." While this is of course of critical importance to IFL, the paper could and should do a better job articulating that the contribution here is on the allocation part of the problem and not the learning part. For example, this could be made clearer in the contributions list in the introduction, and also in Section 4, where the underlying learning algorithm that is (presumably) re-used from other work doesn't appear to be discussed at all.

(-) Related to the point above, it's not clear as to why the authors have chosen the term "Behavior Cloning" for the first method detailed in Section 4.2. In what sense is this related to the term as it is usually understood in the machine learning community (ie, using supervised learning to find a policy that imitates from demonstrations)? This isn't clear to me at all since this appears to again be a solution to the supervisor allocation problem rather than the learning problem. Moreover, having the term "behavior cloning" on the results plots (eg, Figure 2) makes it seems as though _learning_ approaches are being presented rather than _supervisor allocation_ approaches.

(-) Because this paper may serve to define IFL for other researchers in the future, I think it's important that the authors highlight a few other limitations of the way that the problem has been studied here. In particular, it strikes me that this is a type of "federated learning" problem (perhaps something from that community should be cited?), in the context of which the assumptions about communication/synchronization (ie, that timesteps are synchronized across all robots and that they share the same \pi_{\theta_t} for all t) seem like pretty big leaps. For example, the authors cite in the introduction that the IFL problem shows up for deployed fleets of autonomous vehicles. I agree, but certainly these assumptions would be completely invalid in this case. How would that impact the results and conclusions that the authors present here?

**Summary Of Recommendation:**

The paper both helps formalize/advance research in an important problem area and proposes and evaluates an interesting family of algorithms that may help the community make progress in this area. That said, there are several ways in which the paper itself can and should be improved before publication in a venue like CoRL.

POST-DISCUSSION UPDATE: I think the authors have adequately addressed my concerns during the discussion phase, and I think the revised manuscript is greatly improved. As such, I'm increasing my score from "Weak Accept" to "Strong Accept."

---

> ### Author Response · Authors · 2022-08-20
> **Response to Reviewer oMQS**
>
> **We thank Reviewer oMQS for their feedback and positive comment** “The paper articulates well an important problem and introduces a new set of benchmark tasks.” **Below we address each concern. We have attached an updated draft of the paper + appendix to this comment reply with all the changes indicated in the responses below highlighted in blue.**
>
> [T]he Fleet-DAgger approach(es) presented in the paper focus on the particular sub-problem of "supervisor allocation." While this is of course of critical importance to IFL, the paper could and should do a better job articulating that the contribution here is on the allocation part of the problem and not the learning part. For example, this could be made clearer in the contributions list in the introduction, and also in Section 4, where the underlying learning algorithm that is (presumably) re-used from other work doesn't appear to be discussed at all.
>
> **Thank you, this is a great point. A confusing part of our work is that IFL studies supervisor allocation during *learning*, as opposed to prior work that studies supervisor allocation during *execution only*. But, as you point out, the focus is still on the supervisor allocation component. We have revised the contributions list and Section 4 as suggested to more clearly emphasize that the novelty is in the supervisor allocation and that the learning component here is interactive imitation learning from prior work.**
>
> Related to the point above, it's not clear as to why the authors have chosen the term "Behavior Cloning" for the first method detailed in Section 4.2. In what sense is this related to the term as it is usually understood in the machine learning community (ie, using supervised learning to find a policy that imitates from demonstrations)? This isn't clear to me at all since this appears to again be a solution to the supervisor allocation problem rather than the learning problem. Moreover, having the term "behavior cloning" on the results plots (eg, Figure 2) makes it seems as though learning approaches are being presented rather than supervisor allocation approaches.
>
> **Thank you for clarifying this point. As you note, our definition of Behavior Cloning in Section 4.2 was not clear. We have renamed this baseline in Section 4 and the results plots and refined the definition to more clearly explain it. Specifically, this baseline is different from the other algorithms in that it does not receive any teleoperation data or make any policy updates during the 10,000 steps of online training. This is because the robots only receive hard reset interventions (no teleoperation data). In the original draft, we called this baseline Behavior Cloning to indicate that it evaluates how well the robot policy can do when it is only trained on offline human demonstrations via supervised learning. In this sense, it plays a similar role to the BC baseline as defined in interactive imitation learning papers such as ThriftyDAgger [8] and HG-DAgger [49]. As you note, however, the C-prioritization supervisor allocation we add in order to adapt BC to the IFL setting is not relevant to the term “Behavior Cloning.” As such, we changed the name of this baseline to something more appropriate: “Constraint” (i.e., “C” in the C.U.R. naming convention).**
>
> Because this paper may serve to define IFL for other researchers in the future, I think it's important that the authors highlight a few other limitations of the way that the problem has been studied here. In particular, it strikes me that this is a type of "federated learning" problem (perhaps something from that community should be cited?), in the context of which the assumptions about communication/synchronization (ie, that timesteps are synchronized across all robots and that they share the same \pi_{\theta_t} for all t) seem like pretty big leaps. For example, the authors cite in the introduction that the IFL problem shows up for deployed fleets of autonomous vehicles. I agree, but certainly these assumptions would be completely invalid in this case. How would that impact the results and conclusions that the authors present here?
>
> **Thank you, we agree it’s important to highlight limitations. The assumption that the timesteps are synchronous may not hold in scenarios with network latency, communication bottlenecks, etc. We have added a note about this to the Limitations section accordingly as well as a federated learning reference (as they focus on machine learning from the lens of communication efficiency). We also note that one way to make the IFL setting more amenable to high network latency is performing the policy updates to all robots periodically rather than at every time step (e.g., no more than once a day).**

---

### Author Response · Authors · 2022-08-20
**Revision 1 addressing reviewer concerns**

We thank the Reviewers and Area Chair for their helpful feedback and address each concern in a comment reply below. Here we upload an updated draft of the paper + appendix with all the changes indicated in the responses below highlighted in blue.

---

### Author Response · Authors · 2022-08-27
**End of the Author Rebuttal Period**

As the end of the author rebuttal period approaches, we wanted to thank the Reviewers and Area Chair again for their valuable feedback leading to a stronger revised draft. We are happy to see that Reviewer oMQS and 45kd feel that we have addressed their concerns and hope the others feel the same.

---

### Meta-Review · Area_Chair_zeJE · 2022-08-14

**Recommendation:** Accept (Oral)
**Confidence:** 4

**Metareview:**

**Strengths:**

•	Paper is well written, and its contributions are clear.

•	important problem in the robot learning community

•	Interesting benchmark tasks.

•	Open-source codes and the results of simulations

**Weaknesses:**

•	Although contributions are clear, they can be better presented

•	The benefit of the proposed approach over baselines on the physical robot appears marginal.

•	Although the problem of multi-human and the multi-robot setting is novel, the robots share a single policy for the same task.

Post-rebuttal:
Please revise the paper for camera-ready submission according to the rebuttal discussion.


**Best Paper Nomination:**

No

---

> ### Author Response · Authors · 2022-08-20
> **Response to Area Chair zeJE**
>
> **We thank Area Chair zeJE for the feedback and appreciate the AC’s comment** “important problem in the robot learning community”
> **and other positive notes. We have attached an updated draft of the paper + appendix to this comment reply with all the changes indicated in the responses below highlighted in blue. Regarding the concerns, here are specific responses:**
>
> Although contributions are clear, they can be better presented.
>
> **We have revised the contributions and Section 4 to emphasize supervisor allocation more explicitly as Reviewer oMQS suggests.**
>
> The benefit of the proposed approach over baselines on the physical robot appears marginal.
>
> **As Reviewer 45kd requested, we have run new physical experiments with more trials and added error bars to more clearly evaluate and delineate the differences between the approaches. The error bars suggest that the benefits are larger than previously reported.**
>
> Although the problem of multi-human and the multi-robot setting is novel, the robots share a single policy for the same task.
>
> **We acknowledge that our problem setting is restricted to the single-policy case. In the Introduction, we define this as the Fleet setting: as opposed to robot swarms that must coordinate with each other to achieve a common objective, a robot fleet is a set of independent robots simultaneously executing the same control policy in parallel environments. We believe this is an important and understudied setting that is applicable to many real-world commercial and industrial problems (e.g., driverless car fleets and automated warehouses that pool experience and share the same policy), as the reviewers seem to acknowledge with the high Impact scores.**
>
> **Regarding the restriction to the same task, we clarify in the revised version that each MDP can have a different initial state distribution $p_0$. Furthermore, even though the fleet shares the same policy, the “task” can be defined very broadly. For instance, if the fleet learns a goal-conditioned policy where the goal is a language prompt, the same policy can easily be applied to different tasks.**